# Tumoricidal, Temozolomide- and Radiation-Sensitizing Effects of K_Ca_3.1 K^+^ Channel Targeting In Vitro Are Dependent on Glioma Cell Line and Stem Cell Fraction

**DOI:** 10.3390/cancers14246199

**Published:** 2022-12-15

**Authors:** Nicolai Stransky, Katrin Ganser, Ulrike Naumann, Stephan M. Huber, Peter Ruth

**Affiliations:** 1Department of Radiation Oncology, University of Tübingen, 72076 Tübingen, Germany; 2Department of Pharmacology, Toxicology and Clinical Pharmacy, Institute of Pharmacy, University of Tübingen, 72076 Tübingen, Germany; 3Molecular Neurooncology, Hertie Institute for Clinical Brain Research and Center Neurology, University of Tübingen, 72076 Tübingen, Germany

**Keywords:** glioma stem cells, colony formation assay, limited dilution assay, TRAM-34, temozolomide, ionizing radiation

## Abstract

**Simple Summary:**

A potential new treatment for glioma patients is the blockade of K_Ca_3.1 potassium channels. In our study, we performed experiments with the K_Ca_3.1 blocker TRAM-34 in five glioma cell lines. To broaden our findings, effects on cultures enriched in glioma stem cells which are thought to be responsible for treatment failure and relapse were delineated in addition to standard culture conditions. Accordingly, stem-cell enriched cultures were found to be more resistant towards irradiation therapy. Effects of TRAM-34 were dependent on cell line and culture condition and included direct tumoricidal effects, but also temozolomide- and irradiation-sensitizing effects, showing its synergistic potential with current treatment strategies. TRAM-34 effects were mostly found in stem-cell enriched cultures. Overall, our results underline the importance of testing new interventions in several cell lines and different culture conditions to mimic in vivo inter- and intra-tumoral heterogeneity.

**Abstract:**

Reportedly, the intermediate-conductance Ca^2+^-activated potassium channel K_Ca_3.1 contributes to the invasion of glioma cells into healthy brain tissue and resistance to temozolomide and ionizing radiation. Therefore, K_Ca_3.1 has been proposed as a potential target in glioma therapy. The aim of the present study was to assess the variability of the temozolomide- and radiation-sensitizing effects conferred by the K_Ca_3.1 blocking agent TRAM-34 between five different glioma cell lines grown as differentiated bulk tumor cells or under glioma stem cell-enriching conditions. As a result, cultures grown under stem cell-enriching conditions exhibited indeed higher abundances of mRNAs encoding for stem cell markers compared to differentiated bulk tumor cultures. In addition, stem cell enrichment was paralleled by an increased resistance to ionizing radiation in three out of the five glioma cell lines tested. Finally, TRAM-34 led to inconsistent results regarding its tumoricidal but also temozolomide- and radiation-sensitizing effects, which were dependent on both cell line and culture condition. In conclusion, these findings underscore the importance of testing new drug interventions in multiple cell lines and different culture conditions to partially mimic the in vivo inter- and intra-tumor heterogeneity.

## 1. Introduction

Patients with glioblastoma, the most common malignant primary brain tumor in adults [1], exhibit dismal median overall survival times of below two years after multimodal therapy. Standard therapy comprises surgical resection followed by radio-chemotherapy with the DNA-alkylating agent temozolomide and temozolomide maintenance therapy, and, optionally, electrotherapy with tumor-treating fields [2]. One potential new drug target is the intermediate-conductance Ca^2+^-activated potassium channel K_Ca_3.1 (also known as IK, SK4, Gardos Channel or *KCNN4*; for reviews see [3,4]), which may be inhibited by TRAM-34 (1-[(2-Chlorophenyl)diphenylmethyl]-1H-pyrazole, for review see [5]). In the brain, early reports hypothesized K_Ca_3.1 to be solely expressed on malignant cells; however, it is now well-established that normal brain tissues also express K_Ca_3.1, such as brain-resident immune cells [6], astrocytes or neurons [7,8]. Importantly, K_Ca_3.1 expression is reportedly highly upregulated in glioma stem cells [9] and its function has been demonstrated to regulate tumor cell proliferation [10,11] and spreading of glioma cells in the brain [9,12,13,14,15,16]. In addition, blocking K_Ca_3.1 with TRAM-34 reportedly radio-sensitizes glioma cells both in vitro and in vivo [17]. Beyond that, K_Ca_3.1 targeting has been shown to sensitize glioma cells to temozolomide [16]. Both findings underscore potential synergistic effects of K_Ca_3.1 targeting to two main pillars of standard therapy. Finally, high abundance of *K_Ca_3.1* mRNA in glioblastoma resection specimens has been proposed to associate with poor survival times of glioblastoma patients in both the REMBRANDT [14] and TCGA patient cohorts [17], even though this type of analysis suffers from structural weaknesses (as discussed in [18]).

The aim of this work was to define the radio- and chemo-sensitizing effects of pharmacological K_Ca_3.1 targeting in glioma cell lines that differ in genetic background or phenotype. As many earlier reports identified cancer stem cells as more therapy-resistant than “differentiated” cancer cells (for reviews see [19,20]), we compared bulk tumor-differentiating with glioma stem cell-enriching growth conditions in five glioma cell lines.

## 2. Material and Methods

### 2.1. Cell Culture

Three different murine glioma cell lines (SMA-560, SMA-540 and GL-261) and two human glioma cell lines (U-87MG and U-251MG) were studied. SMA-560 and SMA-540 cells originated from the same spontaneous mouse astrocytoma and are, hence, believed to be genetically related [21,22], whereas GL-261 cells were chemically induced by intracranial injection of 3-methylcholantrene in C57BL/6 mice [23]. Both human glioma cell lines were established several decades ago in Sweden [24]. Although several reports describe difficulties tracing back present cultures to original tumor tissues, analyses show glioma origin for both U-87MG and U-251MG cell lines [25,26]. To induce “differentiated” bulk tumor cells, the cell lines were grown in a DMEM medium (ThermoFisher, #41965-039, Austin, TX, USA) supplemented with 10 % fetal bovine serum (FBS) in a 10% CO_2_ atmosphere. For glioblastoma stem cell enriching conditions, cells were cultivated in complete human NeuroCult NS-A Proliferation Medium (including 10 ng/mL rhFGF, 20 ng/mL rhEGF, 2 µg/mL Heparin; STEMCELL Technologies, #05751, #78003, #78006.2, #07980) at 37 °C and 5% CO_2_.

To detach adherent cells or to dissociate cells from spheres, cells were incubated with Trypsin-EDTA 0.05 % (ThermoFisher, #25300054, Austin, TX, USA) for 5–10 min and cell numbers were determined using hemocytometer chips (Neubauer improved). The SMA-540 cells were kindly provided by Hans-Georg Wirsching (Department of Neurology, University Hospital Zurich, Switzerland), and the U-251MG cells were a gift from Dr. Luiz O. Penalva (Graduate School of Biomedical Sciences, UT Health San Antonio, TX, USA).

### 2.2. RNA Isolation and qPCR

mRNA was isolated with NucleoSpin RNA isolation kit (Macherey-Nagel, #740955.250) according to the manufacturer’s instruction. Resulting RNA concentrations were determined with a NanoDrop ND-100 spectrometer. A total of 20 ng of RNA was used for each sample. One step SYBR Green-based reverse transcriptase PCR was accomplished using the 1 Step RT PCR Green ROX L Kit (highQu) following the manufacturer’s instruction. Specific fragments used for murine cell lines were (all Quantitect Primer Assays, Qiagen): *ALDH1A3* (QT01077867), *CXCR4* (QT00249305), *Nestin* (QT00316799), *SOX2* (QT00249347), *KCNN4* (*K_Ca_3.1*, QT00105672). mRNA abundances were normalized to the geometric means of those of the housekeeper genes *GAPDH* (QT01658692) and *PDHB1* (QT00163366). Fragments used for human cell lines were (again all Quantitect Primer Assays, Qiagen): *KCNN4* (*K_Ca_3.1*, QT00003780), *ALDH1A3* (QT00077588), *CXCR4* (QT00223188), *NES* (QT00235781) and *SOX2* (QT00237601). mRNA abundances were normalized to the geometric means of those of the housekeeper genes *GAPDH* (QT01192646) and *ACTB* (QT00095431).

Measurements were conducted on a LightCycler480 (Roche), and crossing point values and melting curves were analyzed using LightCycler 480 software (Roche, version 1.5.0).

### 2.3. Drug Treatment

TRAM-34 (Sigma Aldrich, T6700) was dissolved in DMSO (10 mM stock solution, Sigma Aldrich, D2650) and used at final concentrations of 1 µM (patch-clamp) or 5 µM (all other experiments). The K_Ca_3.1 channel opener 1-EBIO (1-ethyl-2-benzimidazolinone, 200 µM; Sigma Aldrich, SML0034) was diluted from a 20 mM stock solution in DMSO. Temozolomide (Sigma Aldrich, T2577) was dissolved in DMSO (100 mM stock solution) and used at final concentrations of 30 µM. Drugs were added to the cells 1 h before irradiation. Equivalent volumes of DMSO were used as control conditions. Except electrophysiology, all experiments were conducted in a blinded fashion until statistical analysis.

### 2.4. Ionizing Radiation

Irradiation with 6 MV photons of cells at indicated doses was accomplished using a linear accelerator (LINAC SL15, Philips) at a dose rate of 4 Gy/min at room temperature.

### 2.5. Patch-Clamp on-Cell Recording

Macroscopic on-cell (cell-attached) currents from NSC (10–14 d) and DMEM medium-grown SMA-540 mouse glioma cells were recorded in voltage-clamp mode (10 kHz sampling rate) and 3 kHz low-pass-filtered by an EPC-9 patch-clamp amplifier (Heka, Lambrecht, Germany) using Pulse software (Heka) and an ITC-16 Interface (Instrutech, Port Washington, NY, USA). Borosilicate glass pipettes (~5 MΩ pipette resistance; GC150 TF-10, Clark Medical Instruments, Pangbourne, UK) manufactured by a microprocessor-driven DMZ puller (Zeitz, Augsburg, Germany) were used in combination with a STM electrical micromanipulator (Lang, Gießen, Germany). Cells were continuously super-fused at 37 °C with NaCl solution (in mM: 125 NaCl, 32 N-2-hydroxyethylpiperazine-N-2-ethanesulfonic acid (HEPES), 5 KCl, 5 d-glucose, 1 MgCl_2_, 1 CaCl_2_, titrated with NaOH to pH 7.4) additionally containing the K_Ca_3.1 K^+^ channel activator 1-EBIO (0 or 200 µM) and TRAM-34 (0 or 1 µM). The pipette solution contained (in mM) 130 KCl, 32 HEPES, 5 d-glucose, 1 MgCl_2_, 1 CaCl_2_, titrated with KOH to pH 7.4. Currents were elicited by 41 voltage square pulses (700 ms each) from 0 mV holding potential to voltages between −100 mV and +100 mV delivered in 5 mV increments. Clamp voltages refer to the cytosolic face of the plasma membrane and were not corrected for the liquid junction potential between pipette and bath solution. For analysis, macroscopic on-cell currents were averaged between 100 and 700 ms of each voltage sweep. Inward currents are defined as influx of cations into the cells (or efflux of anions out of the cell), depicted as downward deflections of the current tracings, and defined as negative currents in the current voltage relationships. Macroscopic on-cell conductance was calculated for the inward currents between −75 mV and +25 mV clamp voltage. The 1-EBIO-stimulated and TRAM-34-sensitive increase in macroscopic on-cell conductance was used as a measure of functional K_Ca_3.1 channel expression in the plasma membrane.

### 2.6. Clonogenic Survival

Clonogenic survival after drug and/or irradiation treatment was tested using colony formation (attached cells) and limited dilution assays (floating spheres) due to different growth phenotypes of the cell lines (see Appendix A). For delayed plating colony formation assays (SMA-560, SMA-540 and GL-261 cells), 600 cells were seeded per well in 6-well plates in a drug-free medium 24 h after irradiation (0–6 Gy). 60 min prior to 24 h after irradiation, cells were incubated with TRAM-34 (0 or 5 µM) and temozolomide (0 or 30 µM). For pre-plating colony formation assays (U-87MG and U-251MG cells), 300 cells were seeded per well. After 4 h of incubation period, cells were incubated with TRAM-34 (0 or 5 µM) and temozolomide (0 or 30 µM). Irradiation treatment was conducted 1 h after TRAM-34 or temozolomide were added. After 5–7 (SMA-560, SMA-540 and U-87MG), 11 (U-251MG) or 13–14 (GL-261) days, cells were fixated with 4.5% formaldehyde and stained with 0.05 % Coomassie blue. Resulting colonies, defined as cell clusters consisting of at least 50 cells, were counted. Plating efficiency was calculated by dividing the number of colonies by the number of plated cells. Survival fractions were calculated by dividing the plating efficiencies by the respective plating efficiency at 0 Gy of each treatment arm. All experiments consist of 3 experimental and 3-6 observational units each (for definitions of experimental and observational unit, see [27]).

For limited dilution assays, singularized cells were serially 1:2 diluted in 96-well plates resulting in cell numbers between 2048 and 1 cell per well. A total of 24 h after cell plating, cells were pretreated (60 min) and continuously post-treated before and after irradiation (0–8 Gy) with TRAM-34 (0 or 5 µM) and temozolomide (0 or 30 µM). After a further 5–7 (SMA-560, U-87MG), 11 (U-251MG) or 14 days (SMA-540, GL-261), minimal cell number to retain the culture was determined. Plating efficiency was defined as the reciprocal value of this minimal cell number. The survival fraction was calculated as mentioned above. All experiments consist of three experimental and four observational units each. Fitted curves were calculated according to the linear quadratic model [28] with the following equation:$$Survival\;Fraction=e^{-\left(\alpha \cdot D+\beta \cdot D^{2}\right)}$$
with *D* = radiation dose, and *α* and *β* = cell-type specific parameters.

## 3. Results

To account for genetic, but also phenotypical differences, three different murine and two human glioma cell lines were grown in two different culture media, DMEM supplemented with 10% fetal bovine serum that should “differentiate” glioblastoma “bulk” cells [29,30], and a neural stem cell-enriching/inducing NSC medium. To assess the influence of these two incubation conditions on stem cell properties, the mRNA abundance of four established stem cell markers, *ALDH1A3* [31], *CXCR4* [32], *Nestin* [33] and *SOX2* [34] was determined (Figure 1A–D). All but U-87MG (SMA-560, SMA-540, GL-261 and U-251MG) showed statistically significant increases in mRNA abundance in one or more of the stem cell markers when grown in NSC medium (see Figure 1). For example, mRNA abundance of *ALDH1A3* was more than doubled (*p* = 0.0183), while mRNA abundance of *CXCR4* was more than 100× greater when SMA-540 cells were grown in NSC compared to standard DMEM culture. U-87MG cells were the only cell line for which no statistically significant increase in any of the four analyzed stem cell markers was observed, even though *Nestin* was upregulated on a low basal level (*p* = 0.0883). Combined, these data suggest that stem cell-enriching culture conditions indeed increased the expression of stem cell markers in all but one cell line studied.

Next, the effect of this stem cell enrichment on survival fractions after irradiation was analyzed. For “differentiated” cells grown in DMEM culture medium, colony formation assays (CFA) were performed. Since the cell lines formed floating or attached spheroids when cultured in NSC medium (Appendix A), we used limited dilution assays (LDA) for NSC-grown cells. To exclude LDA vs. CFA inter-assay differences, SMA-560 cells, grown in DMEM medium, were tested both in CFAs and LDAs, which resulted in only negligible differences of clonogenicity (see Appendix A) indicating inter-assay concordance.

Comparison of the two culturing media suggested that stem cell-enriching conditions led to higher survival fractions in irradiated SMA-560 and SMA-540 cells, as compared to “differentiated” DMEM-grown cells (Figure 2A,B). No differences were observed with GL-261 cells (Figure 2C). In contrast, U-87MG cells cultured in DMEM medium exhibited higher numerical survival fractions, even though data largely overlapped between both culture media (Figure 2D). Last, human U-251MG cells showed a doubling in survival fraction when grown in stem cell-enriching NSC medium (*p* = 0.056, Figure 2E).

To test the influence of stem-cell enriching culture conditions on *K_Ca_3.1* (*KCNN4*) mRNA, we performed real-time RT-PCR analyses for all five cell lines under both culture conditions. SMA-560 cells expressed *K_Ca_3.1* independently of culture conditions to a moderate-to-high extent (Figure 3A). In SMA-540 cells, K_Ca_3.1 mRNA abundance was increased by twofold under stem cell enriching as compared to DMEM culture conditions, even though statistical significance was not reached (Figure 3B). Both murine GL-261 (Figure 3C) and human U-87MG cells did express *K_Ca_3.1* to a much lower extent, and this expression was not influenced by the culture condition (Figure 3D). U-251MG cells also expressed *K_Ca_3.1* on a low-to-moderate level; however, its expression almost doubled when cultured in NSC medium (Figure 3E). Irradiation of cells had no further systematic influence on mRNA abundances of stem cell-associated genes or *K_Ca_3.1* (Appendix A) in all three murine cell lines.

To confirm functional expression of K_Ca_3.1 K^+^ channels in the plasma membrane of SMA-540 murine glioma cells and to identify its dependence on the culture conditions, macroscopic on-cell (cell-attached) currents as obtained with patch-clamp recordings in voltage-clamp mode were compared between cells grown in NSC medium and continuously DMEM medium-cultured sub-confluent monolayers. Macroscopic on-cell currents were recorded with KCl pipette and NaCl bath solution at clamp voltages between −100 and +100 mV. Maximal K_Ca_3.1 K^+^ channel activity was first induced by bath super-fusion with the K_Ca_3.1 opener 1-EBIO (200 µM) and then blocked by co-super-fusion of the K_Ca_3.1 blocker TRAM-34 (1 µM). The TRAM-34-sensitive current fraction was analyzed as a measure of functional K_Ca_3.1 surface expression (Figure 4A,B).

In on-cell mode, the physiological membrane potential also applies to the electrically sealed membrane patch and contributes additively to the transmembrane voltage of the recorded membrane patch on top of the clamp voltage. With KCl pipette solution and with an assumed high intracellular K^+^ concentration, the *E*_K_ electrochemical equilibrium potential for potassium across the recorded membrane patch can be expected to be around 0 mV transmembrane voltage, i.e., when the positive clamp voltage zeroes the negative physiological membrane potential. Since (further) activation of the K_Ca_3.1 K^+^ conductance by 1-EBIO is expected to hyperpolarize the membrane potential and K_Ca_3.1 blockage by TRAM-34 to depolarize it, K_Ca_3.1 activation and blockage should be paralleled by an increase and decrease, respectively, of the positive clamp voltage that is required to zero the physiological membrane potential. Assuming finally that the K^+^ conductance is the largest conductance fraction in the plasma membrane of glioma cells and this fraction largely determines the membrane potential, the reversal potential (V*_rev_*) of the recorded macroscopic current should to some extent represent *E*_K_ and thereby the negative value of the membrane potential.

In our experiments on SMA-540 cells, 1-EBIO induced an increase in the macroscopic on-cell inward currents and TRAM-34 the blockage of these currents (Figure 4B). Notably, on average, the 1-EBIO-induced increase in macroscopic on-cell inward current and TRAM-34-sensitive current fractions seemed to be larger in SMA-540 cells grown in NSC medium than in DMEM medium-cultivated cells (Figure 4C,D). As a matter of fact, a significant and almost significant higher number of NSC medium-grown and 1-EBIO-treated cells exhibited a large inward conductance and a high TRAM-34-sensitive conductance fraction, respectively, than 1-EBIO-treated DMEM medium cultivated cells. Moreover, and in accordance with the above-mentioned assumptions, 1-EBIO induced a right shift (i.e., an increase) and TRAM-34 an left shift (i.e., decrease), respectively, in V_rev_ of the macroscopic on-cell current especially in NSC medium-grown SMA-540 cells (Figure 4C,F). In two-by-two contingency plots (Figure 4G), however, number of cells with high V_rev_ and large TRAM-34-induced V_rev_ decline were not significantly different between both culture conditions. Combined, this data indicate functional expression of K_Ca_3.1 channels in SMA-540 murine glioma cells and suggest upregulation of K_Ca_3.1 in the plasma membrane upon transferring SMA-540 from DMEM in NSC medium.

To test for the functional significance of K_Ca_3.1 on clonogenic survival of DMEM cultured and stem-cell enriched (NSC culture) glioma cells after irradiation (0–8 Gy) and/or chemotherapy, we applied the K_Ca_3.1 blocker TRAM-34 (0 or 5 µM) in combination with temozolomide (0 or 30 µM). As shown in Figure 5A–C, temozolomide or TRAM-34 alone hardly affected plating efficiency of SMA-560 cells in colony formation assay or limited dilution assay. TRAM-34, however, sensitized SMA-560 cells to temozolomide when grown in DMEM medium, while this trend was only numerical when grown in NSC medium. Furthermore, in DMEM-grown irradiated (0–6 Gy) SMA-560 cells, neither TRAM-34 nor temozolomide, nor their combination, decreased survival fraction in colony formation assay (Figure 5D), suggesting no radio-sensitizing action of both drugs alone or their combination in “differentiated” SMA-560 cells. In limited dilution assay with stem cell-enriched NSC-grown SMA-560 cells, temozolomide showed a trend (*p* = 0.056) towards radio-sensitization of the cells at high irradiation doses (Figure 5D). TRAM-34 alone, in contrast, did not exhibit any effect on survival fraction of irradiated stem cell-enriched SMA-560 cells (Figure 5E).

In SMA-540 cells, neither TRAM-34 nor temozolomide, nor their combination, decreased plating efficiency in both culture conditions (Figure 6A–C). However, while neither TRAM-34 nor TMZ affected survival fractions after irradiation in DMEM-cultured SMA-540 cells (Figure 6D), both agents (and their combination) did significantly reduce survival fractions at high irradiation doses in stem-cell enriched SMA-540 cells (Figure 6E).

In contrast to SMA-540 and SMA-560 cells, GL-261 cells were highly temozolomide-sensitive irrespective of culture conditions. Temozolomide reduced plating efficiency in colony formation assay and limited dilution assay by approximately 85% and 97%, respectively (Figure 7A–C; *p* < 0.0001). Further reductions by the addition of TRAM-34 were small and only numerical (Figure 7B,C). Due to temozolomide’s large effect on plating efficiency on its own, we subsequently only analyzed TRAM-34′s effects on survival fraction after irradiation. TRAM-34 had no effect on survival fraction in DMEM-grown “differentiated” GL-261 cells (Figure 7D). In stem cell-enriched culture, TRAM-34 treatment showed a trend towards higher survival fractions, i.e., towards an increased radioresistance (Figure 7E).

Next, we did not identify changes in plating efficiency for human U-87MG cells cultured in DMEM medium for any treatment group (Figure 8A,B). Interestingly, NSC cultured U-87MG showed an 80% reduction in plating efficiency when incubated with TRAM-34, and near total reductions when TRAM-34 and TMZ were applied concomitantly (Figure 8C). We observed only small effects of any treatment on survival fraction after irradiation in DMEM cultured U-87MG cells, with only numerical reductions of the survival fraction at 3 Gy in the TMZ and TRAM-34 + TMZ treatment group. Due to the large effects of TRAM-34 and TRAM-34 + TMZ on U-87MG cells cultured in NSC medium alone, only TMZ’s effect was analyzed regarding its effect on radiation sensitivity, showing ambiguous results depending on the irradiation dose (Figure 8D,E).

Last, we analyzed the human U-251MG cell line, which exhibited a large sensitivity towards TMZ with large reductions in plating efficiency in both culture conditions (Figure 9A–C). Moreover, TRAM-34 reduced plating efficiency in limited dilution assay by approximately 40%, even though statistical significance was not reached (Figure 9C; *p* = 0.0649). Due to the large effects of TMZ on its own, only TRAM-34 treatment was subsequently analyzed regarding its radio-sensitization effects. Overall, only small effects of TRAM-34 trending towards reduced survival fractions after irradiation were found (Figure 9D,E).

## 4. Discussion

The present study showed that radio-sensitivity was influenced by culture conditions. SMA-560, SMA-540 and (to a smaller degree) U-251MG cells were more radio-resistant when cultured in stem cell-enriching NSC medium as compared to the “bulk” glioma DMEM/FBS cell cultures. Culture medium-induced changes of radiation sensitivity in GL-261 cells and U-87MG were only small (see Figure 2). Notably, U-87MG cells were the only cell line studied which did not show a clear induction of any of the four tested stem cell markers when grown in NSC medium. The increased radioresistance in stem cell-enriched cultures is in line with previous reports, showing increased radioresistance of cancer stem cells as compared to “differentiated bulk” tumor cells, which is most probably due to upregulation of repair mechanisms, increased oxidative defense and/or activation of pro-survival pathways in cancer stem cells (as reviewed in [20]).

The main finding of the present study is that tumoricidal, but also radio- and temozolomide-sensitizing effects of the K_Ca_3.1 inhibitor TRAM-34 varied considerably between glioma cell lines and culture conditions. Specifically, TRAM-34 had radio-sensitizing effects in stem cell-enriched but not in “differentiated” SMA-540 cells (see Figure 6D,E). While TRAM-34 had no effect on clonogenic survival in DMEM-cultured U-87MG cells, plating efficiency was reduced by 80% when applied to NSC-cultured U-87MG cells (see Figure 8B,C). A similar trend was found in U-251MG; however, TRAM-34′s effect in NSC medium did not quite reach statistical significance (see Figure 9B,C; *p* = 0.0649). Furthermore, no effect of TRAM-34 was seen in GL-261 cells, arguably due to its low K_Ca_3.1 expression compared to the other cell lines (see Figure 3 and Figure 7). This contrasts a previous study [16], which reported both direct tumoricidal but also TMZ-sensitizing effects of TRAM-34 in GL-261 cells. Notably, D’Alessandro et al. [16] applied 5 µM TRAM-34 in DMEM medium containing 1 % FBS for TRAM-34 single and temozolomide co-treatment, while our experiments on “bulk-differentiated” GL-261 cells were conducted with 5 µM TRAM-34 in DMEM medium containing 10% FBS (see Section 2).

In theory, differences in free TRAM-34 concentration may explain intra-cell line differences between both culture conditions, and the differences between this and D’Alessandro et al.’s paper: Given the reported high plasma protein binding rates (around 98% in rat plasma) of TRAM-34 [35], it is justified to assume higher free TRAM-34 concentrations in culture media with less or no FBS, such as NSC. Nevertheless, IC_50_ values for TRAM-34 are reportedly around 20 nM [36], which should ensure quantitative blockage of K_Ca_3.1 even when used in cell culture medium containing 10% FBS. To explore this possibility further, we conducted CFA using DMEM + 1% FBS with U-87MG cells, which showed the largest reduction in plating efficiency with TRAM-34 treatment in NSC medium (see Figure 8B). While colony formation was slower, TRAM-34 treatment did not lead to reductions in plating efficiency and resembled findings as when grown in DMEM + 10% FBS, rendering this hypothesis unlikely (Appendix A).

Alternatively, this high variability might result from cell line- and culture condition-dependent differences in *K_Ca_3.1* expression. While *K_Ca_3.1* mRNA abundance did not differ in SMA-560 (high *K_Ca_3.1* expression), GL-261 (low *K_Ca_3.1* expression) and U-87MG (low *K_Ca_3.1* expression) cells between both culture conditions, SMA-540 cells (high *K_Ca_3.1* expression; *p* = 0.154) and U-251MG (low *K_Ca_3.1* expression) showed an increase in K_Ca_3.1 mRNA abundance with stem cell enrichment in NSC medium (see Figure 3), which was also seen functionally in SMA-540 cells with patch-clamp recording (see Figure 4). This might explain why a radio-sensitizing effect of TRAM-34 alone and an additive radio-sensitizing effect of TRAM-34 combined with temozolomide treatment was only apparent in glioma stem cell-enriched SMA-540 cells (see Figure 6D–E) which was the culture with the highest *K_Ca_3.1* mRNA abundance (see Figure 3B). In contrast to this line of argumentation, the large tumoricidal effect of TRAM-34 in NSC-cultured U-87MG cells was not observed in DMEM culture, even though the expression of K_Ca_3.1 did not differ between both culture conditions and was generally low compared to SMA-540 cells (see Figure 3D and Figure 8B,C).

Temozolomide did not reduce the clonogenic survival in SMA-560, SMA-540 and U-87MG cells or only did so to a small, statistically non-significant amount (see Figure 5B,C, Figure 6B,C and Figure 8B,C). Speculatively, this may be due to high expression of the temozolomide resistance gene O6-methylguanine-DNA-methyltransferase (MGMT), or (as is the case for U-87MG with a methylated and hence low MGMT expression status [37]) result from a reduced expression of mismatch repair proteins [38,39], or an enhanced expression of DHC2, which interferes with nuclear transportation of DNA repair proteins [40]. This is contrasted by large and culture condition-independent effects of TMZ in GL-261 and U-251MG cells (see Figure 7B,C and Figure 9B,C).

In contrast to the present study, previous work of our group and others have consistently disclosed a radioprotective function of K_Ca_3.1. In particular, TRAM-34 treatment has been shown to radio-sensitize human glioblastoma cell lines (T98G and U87MG, both cultured in RPMI medium supplemented with 10% FCS and using higher TRAM-34 concentrations of up to 10 µmol/L). Strikingly, knockdown of *K_Ca_3.1* by RNA interference in T98G cells mimicked the radio-sensitizing TRAM-34 effect and concomitantly abolished the sensitivity to TRAM-34 [17]. Vice versa, downregulation of the stem cell marker *musashi-1* in U251 human glioblastoma cells reportedly upregulates *K_Ca_3.1* and induces TRAM-34-sensitive radioresistance [41]. Likewise, genetic knockdown of *K_Ca_3.1* radio-sensitizes murine breast cancer cells with loss of TRAM-34-sensitive radioresistance [42].

Combined with the data of the present study, this might suggest that K_Ca_3.1 targeting with TRAM-34 may sensitize glioma cells to radiation (or temozolomide) only under specific culture conditions and in selected glioma cell lines/cultures, speculatively preferentially in stem cell-enriching (serum-free) culture conditions. In our previous work on patient-derived primary glioblastoma stem cell cultures, *K_Ca_3.1* expression varied considerably between individual tumors and was associated with mesenchymal subpopulations of stem cells [3,43], which are especially radiation-resistant [44] and may, hence, represent the subpopulation of glioblastoma cells to test further anti-K_Ca_3.1 therapies in.

Importantly, several groups identified also anti-migratory or anti-invasive effects of anti-K_Ca_3.1 treatment [9,12,13,14,15,16], properties which were not part of our present assessment. Moreover, other authors found the immune constitution of animals to be a better predictor for response to radiation treatment than intrinsic radiosensitivity of cancer cells [45]. This might be especially interesting in light of the findings of Grimaldi et al. [46], demonstrating anti-immunosuppressive effects of TRAM-34 in glioma-infiltrating microglia/macrophages, which led to decreased tumor volumes in a glioma mouse model.

*Limitations.* There are several limitations, potentially challenging the generalizability of the present study. First, NSC medium was originally developed for culturing human stem cells. However, as shown in Figure 1, expression of stem cell associated genes was also increased in all three murine cell lines. Second, while we did show inter-assay concordance of CFAs and LDAs, incubation times were chosen based on proliferation rate of each cell line in each culture medium. This led to different incubation times of SMA-540 cells in CFA and LDA (7 days versus 14 days), which may complicate direct comparisons of radiosensitivity between DMEM and NSC cultures of SMA-540 cells. Last, determining drug concentrations is challenging. While there is evidence that TMZ may reach brain tissue concentrations of up to 30 µM in human patients [47,48], no human pharmacokinetic data are available for TRAM-34. Experiments in rats identified peak concentration levels of 2.5 µmol/l in brain tissue after intraperitoneal injections [35], while other authors found concentrations of up to 1.3 µmol/L in mice brain tissue [13].

## 5. Conclusions

We identified tumoricidal, TMZ- and radiation-sensitizing effects of pharmacological K_Ca_3.1 targeting by TRAM-34 in our tested glioma cell lines. However, these effects were not only cell line-specific, but also dependent on the culture conditions used. Notably, TRAM-34 was especially effective against stem-cell enriched cell cultures, which are generally thought of as responsible for therapy resistance and tumor relapse. This underpins the importance of testing new drug targets in various cell lines and different culture conditions, to mimic intra- and inter-tumor heterogeneity in glioblastoma patients at least partially [49,50].

## Figures and Tables

**Figure 1 cancers-14-06199-f001:**
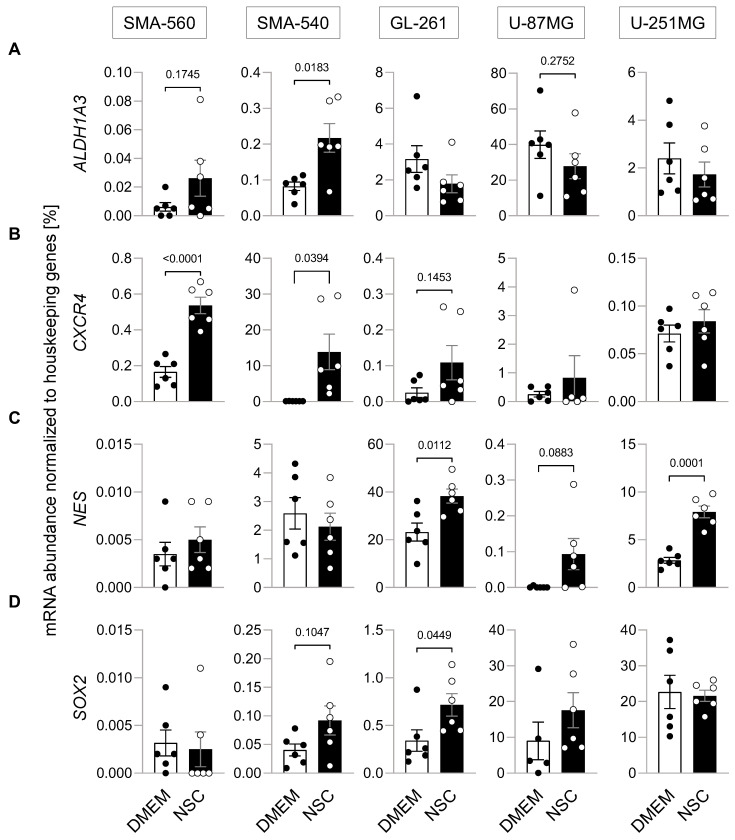
Enrichment of the glioblastoma stem cell fraction. (**A**–**D**). Culturing murine SMA-560, SMA-540 and GL-261 glioma cells and human U-251MG glioma cells in NSC medium (open circles) increased housekeeper-normalized mRNA abundances of stem cell markers as compared to DMEM medium (closed circles; in addition to the individual values, means ± standard error of three experimental and two observational units each are depicted). The mRNA abundances of four different stem cell markers, (**A**) *ALDH1A3*, (**B**) *CXCR4*, (**C**) *Nestin* and (**D**) *SOX2* were determined by real-time RT-PCR. Numbers indicate error probability (*p* values) as calculated by Welch-corrected two-tailed *t*-test.

**Figure 2 cancers-14-06199-f002:**
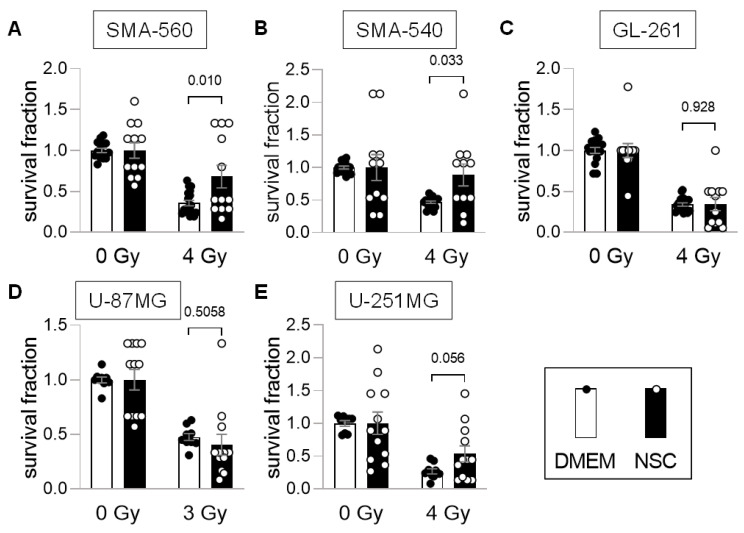
SMA-560, SMA-540 and U-251MG stem cell-enriched cultures are more radio-resistant than “differentiated” cultures. (**A**–**E**). Compared to standard culturing medium DMEM (closed circles; open bars), stem cell-inducing culture conditions with NSC medium (open circles; closed bars) increased clonogenic survival in (**A**) SMA-560, (**B**) SMA-540 and (**E**) (statistically non-significantly) U-251MG cells after irradiation. No statistically significant difference was observed in (**C**) GL-261 and (**D**) U-87MG cells (means ± SE of 3 experimental units with 4–6 observational units each). Numbers indicate error probabilities (*p* values) as calculated by Welch-corrected two-tailed *t*-test.

**Figure 3 cancers-14-06199-f003:**
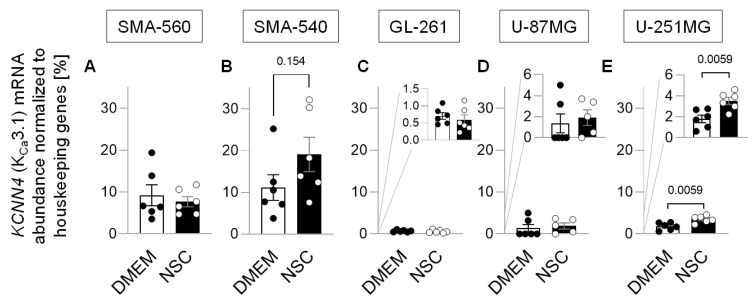
Effect of the enrichment of the glioblastoma stem cell fraction on *K_Ca_3.1* mRNA abundance. (**A**–**C**). Housekeeper-normalized mRNA abundance of *K_Ca_3.1* (*KCNN4*) in SMA-560 (**A**), SMA-540 (**B**), GL-261 (**C**), U-87 MG (**D**) and U-251MG (**E**) cells in both DMEM (closed circles) and NSC (open circles) medium (in addition to the individual values, means ± standard error of three experimental and two observational units each are depicted; numbers in (**B**,**E**) indicate error probability (*p* value) as calculated by Welch-corrected two-tailed *t*-test.

**Figure 4 cancers-14-06199-f004:**
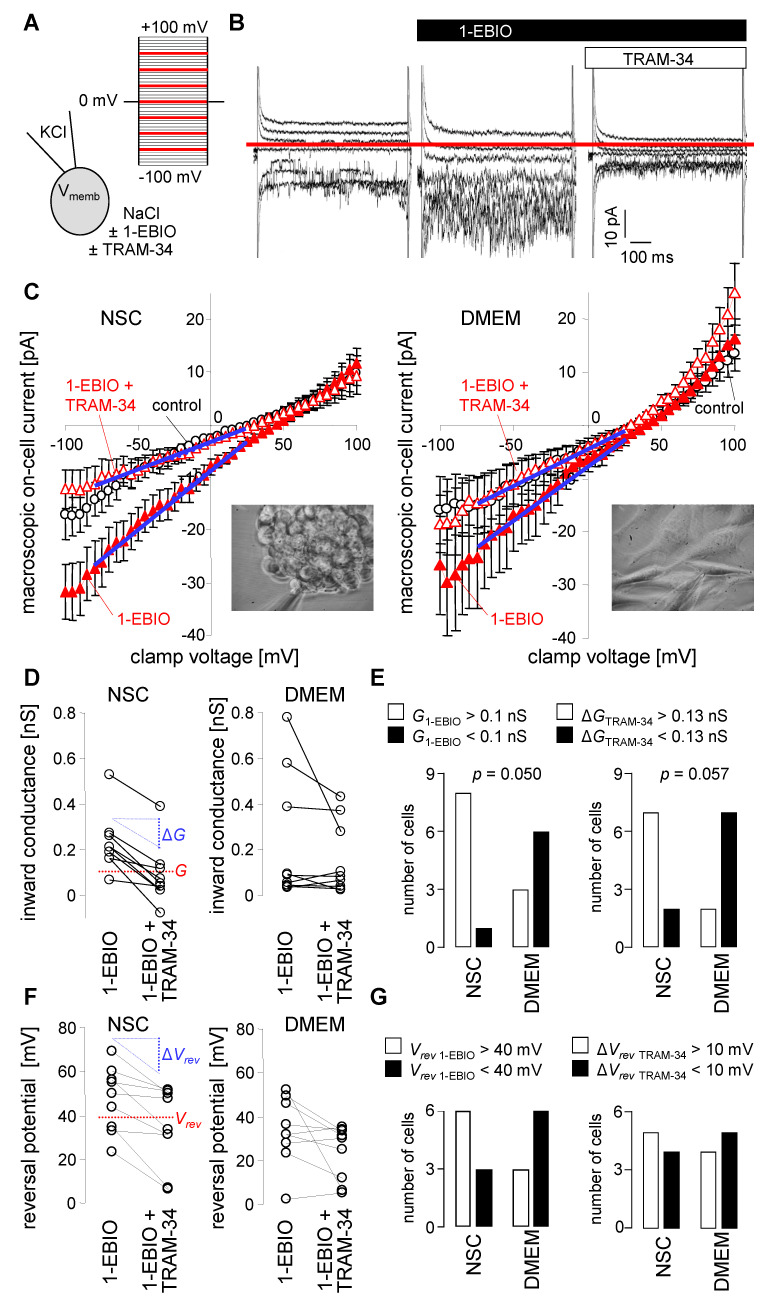
NSC culture conditions induce upregulation of K_Ca_3.1 K^+^ currents in the plasma membrane of SMA-540 mouse glioma cells (**A**,**B**). Ionic composition with channel modulators of pipette and bath solution and applied voltage pulse protocol (**A**) used to record macroscopic on-cell (cell-attached) currents from an NSC-grown (10–14 d) SMA-540 cell. On-cell current tracings depicted in (**B**) were obtained before (**left**), during bath application of the K_Ca_3.1 K^+^ channel opener 1-EBIO (200 µM) alone (**middle**) or in combination with the K_Ca_3.1 inhibitor TRAM-34 (1 µM, **right**). For better readability, only current tracings elicited by voltage square pulses to −75, −50, −25, 0, +25, +50, and +75 mV are shown; red line in (**B**) indicates 0 pA (**C**). Dependence of macroscopic on-cell currents on clamp voltage in NSC- (**left**) and DMEM-grown (**right**) SMA-540 cells recorded before (control, open black circles), during bath application of 1-EBIO (closed red triangles) and co-application of 1-EBIO and Tram-34 (open red triangles). Data are means ± SE, *n* = 9; the inserts in the lower right corner of the plots show light micrographs of a SMA-540 spheroid (**left**) and a SMA-540 monolayer (**right**) during patch-clamp recording (**D**). Paired conductances as calculated for the inward currents from the data summarized in (**C**) in NSC- (**left**) and DMEM-grown (**right**) SMA-540 cells recorded successively with 1-EBIO and 1-EBIO/TRAM-34 in the bath solution. Given are individual values determined for that range of clamp voltage indicated in (**C**) by blue lines (**E**). Two-by-two contingency plots showing the number of NSC- and DMEM-grown SMA-540 cells with an on-cell inward conductance in 1-EBIO-containing bath solution (*G*_1-EBIO_) of above (closed columns) and below (open columns) 0.1 nS (**left**) as well as with a TRAM-34-induced inward conductance decline (Δ*G*_TRAM-34_) of above (closed columns) and below (open columns) 0.13 nS (**right**). Data are from (**D**), indicated *p* values refer to the difference between the culture conditions and were calculated with chi square test (**F**). Paired reversal potentials (*V*_rev_s, individual values) as given for the data summarized in (**C**) in NSC- (**left**) and DMEM-grown (**right**) SMA-540 cells recorded successively with 1-EBIO and 1-EBIO/TRAM-34 in the bath solution (**G**). Two-by-two contingency plots showing the number of NSC- and DMEM-grown cells with a *V*_rev_ in 1-EBIO-containig bath solution (*V*_rev 1-EBIO)_ of above (closed columns) and below (open columns) +40 mV (**left**) as well as with a TRAM-34-induced drop in *V*_rev_ (Δ*V*_rev TRAM-34_) by above (closed columns) and below (open columns) 10 mV (**right**).

**Figure 5 cancers-14-06199-f005:**
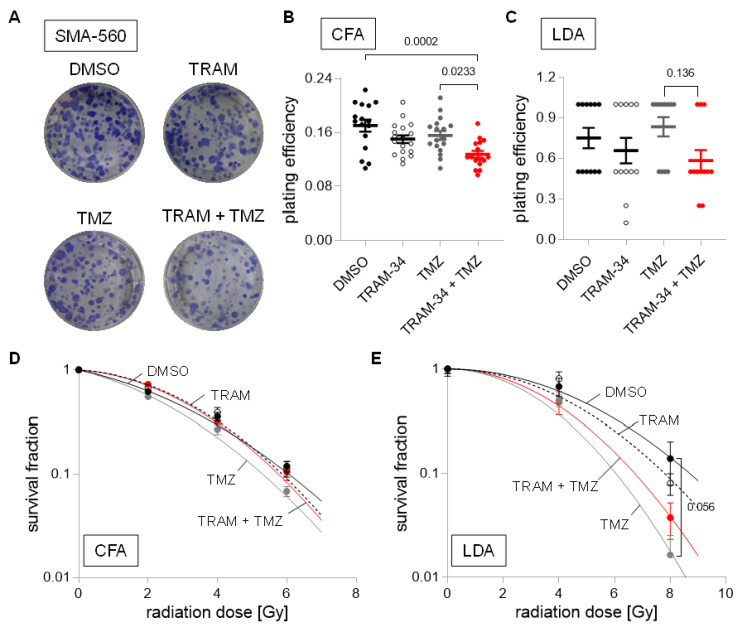
Cell culture condition-dependent effects of K_Ca_3.1 blockade on sensitivity to irradiation and temozolomide in SMA-560 cells (**A**–**C**). Representative images of colony formation (**A**) and plating efficiencies (**B**) of DMEM-grown or (**C**) NSC-grown cells after irradiation with 0 Gy and co-treatment with vehicle (DMSO, closed black circles), TRAM-34 (5 µM, open circles), temozolomide (TMZ, 30 µM, closed grey circles), or TRAM-34 (5 µM) together with temozolomide (TMZ, 30 µM, closed red circles) (**D**,**E**) Survival fractions of irradiated (0–8 Gy) cells co-treated with vehicle (DMSO, closed black circles), TRAM-34 (5 µM, open circles), temozolomide (TMZ, 30 µM, closed grey circles), or TRAM-34 (5 µM) together with temozolomide (TMZ, 30 µM, closed red circles) as determined by colony formation assay for DMEM-grown (**D**) and limited dilution assay for NSC-grown SMA-560 cells (**E**). Data are individual values in (**B**,**C**) and mean ± standard error in (**B**–**E**) of three experimental units with four to six observational units each. Survival fraction curves were fitted according to the linear quadratic model and given as follows: DMSO, solid black line; TRAM-34, dashed line; temozolomide, solid grey line; TRAM-34 and TMZ, solid red line. Numbers in (**B**,**C**,**E**) indicate error probabilities (*p* values) as calculated with one-way ANOVA and Tukey multiple comparison test.

**Figure 6 cancers-14-06199-f006:**
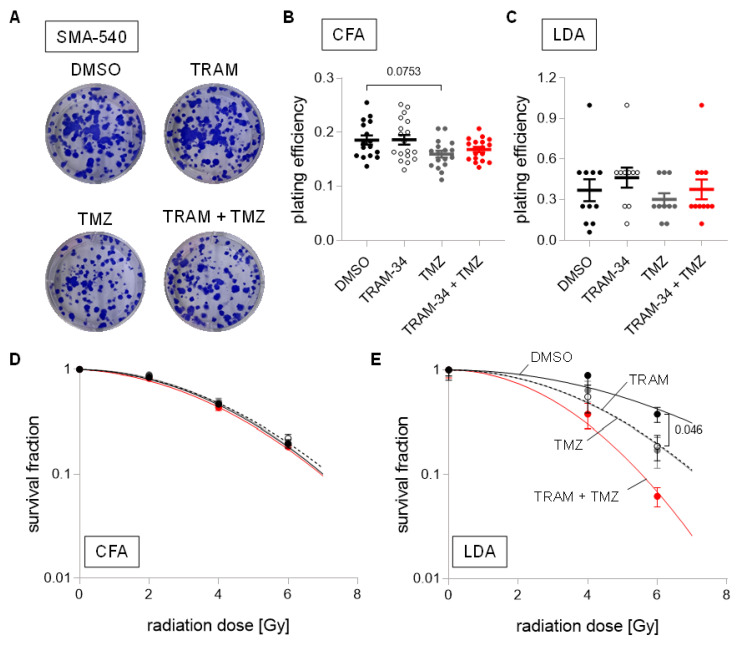
Cell culture condition-dependent effects of K_Ca_3.1 blockade on sensitivity to irradiation and temozolomide in SMA-540 cells. (**A**–**C**). Representative images of colony formation (**A**) and plating efficiencies (**B**) of DMEM-grown or (**C**) NSC-grown cells after irradiation with 0 Gy and co-treatment with vehicle (DMSO, closed black circles), TRAM-34 (5 µM, open circles), temozolomide (TMZ, 30 µM, closed grey circles), or TRAM-34 (5 µM) together with temozolomide (TMZ, 30 µM, closed red circles). (**D**,**E**). Survival fractions of irradiated (0–6 Gy) cells co-treated with vehicle (DMSO, closed black circles), TRAM-34 (5 µM, open circles), temozolomide (TMZ, 30 µM, closed grey circles), or TRAM-34 (5 µM) together with temozolomide (TMZ, 30 µM, closed red circles) as determined by colony formation assay for DMEM-grown (**D**) and limited dilution assay for NSC-grown SMA-540 cells (**E**). Data are individual values in (**B**,**C**) and mean ± standard error in (**B**–**E**) of three experimental units with four to six observational units each. Survival fraction curves were fitted according to the linear quadratic model and given as follows: DMSO, solid black line; TRAM-34, dashed line; temozolomide, solid grey line; TRAM-34 and TMZ, solid red line. Numbers in (**B**,**E**) indicate error probabilities (*p* values) as calculated with one-way ANOVA and Tukey multiple comparison test.

**Figure 7 cancers-14-06199-f007:**
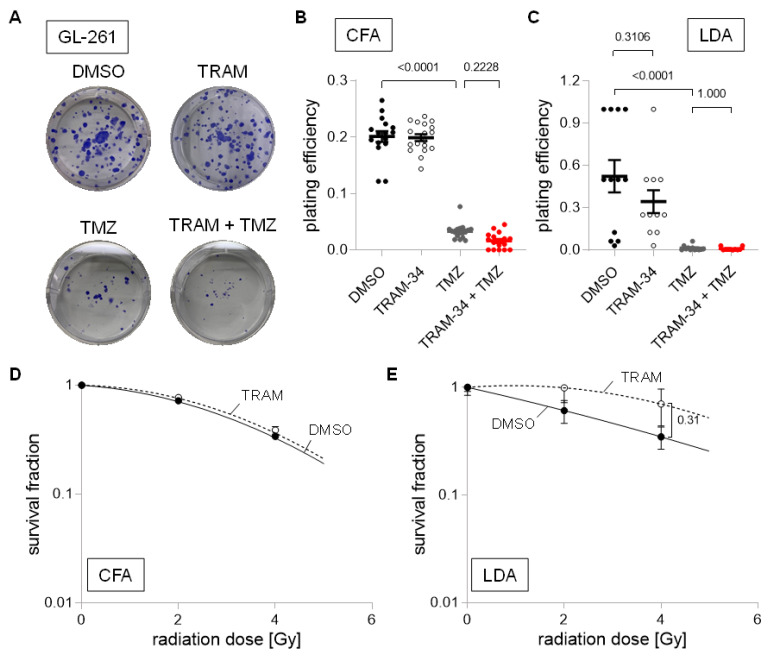
Cell culture condition-dependent effects of K_Ca_3.1 blockade on sensitivity to irradiation and temozolomide in GL-261 cells. (**A**–**C**). Representative images of colony formation (**A**) and plating efficiencies (**B**) of DMEM-grown or (**C**) NSC-grown cells after irradiation with 0 Gy and co-treatment with vehicle (DMSO, closed black circles), TRAM-34 (5 µM, open circles), temozolomide (TMZ, 30 µM, closed grey circles), or TRAM-34 (5 µM) together with temozolomide (TMZ, 30 µM, closed red circles). (**D**,**E**). Survival fractions of irradiated (0–4 Gy) cells co-treated with vehicle (DMSO, closed black circles) or TRAM-34 (5 µM, open circles) as determined by colony formation assay for DMEM-grown (**D**) and limited dilution assay for NSC-grown GL-261 cells (**E**). Data are individual values in (**B**,**C**) and mean ± standard error in (**B**–**E**) of three experimental units with four to six observational units each. Survival fraction curves were fitted according to the linear quadratic model and given as follows: DMSO, solid black line; TRAM-34, dashed line. Numbers in (**B**,**C**,**E**) indicate error probabilities (*p* values) as calculated with one-way ANOVA and Tukey multiple comparison test.

**Figure 8 cancers-14-06199-f008:**
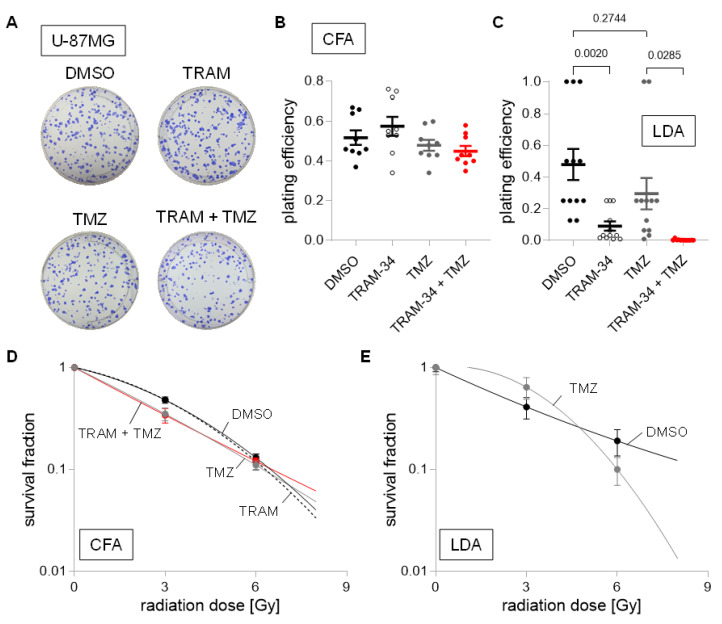
Cell culture condition-dependent effects of K_Ca_3.1 blockade on sensitivity to irradiation and temozolomide in U-87MG cells. (**A**–**C**). Representative images of colony formation (**A**) and plating efficiencies (**B**) of DMEM-grown or (**C**) NSC-grown cells after irradiation with 0 Gy and co-treatment with vehicle (DMSO, closed black circles), TRAM-34 (5 µM, open circles), temozolomide (TMZ, 30 µM, closed grey circles), or TRAM-34 (5 µM) together with temozolomide (TMZ, 30 µM, closed red circles). (**D**,**E**). Survival fractions of irradiated (0–6 Gy) cells co-treated with vehicle (DMSO, closed black circles), TRAM-34 (5 µM, open circles), temozolomide (TMZ, 30 µM, closed grey circles), or TRAM-34 (5 µM) together with temozolomide (TMZ, 30 µM, closed red circles) as determined by colony formation assay for DMEM-grown (**D**) and limited dilution assay for NSC-grown U-87MG cells (**E**). Data are individual values in (**B**,**C**) and mean ± standard error in (**B**–**E**) of three experimental units with three to four observational units each. Survival fraction curves were fitted according to the linear quadratic model and given as follows: DMSO, solid black line; TRAM-34, dashed line; temozolomide, solid grey line; TRAM-34 and TMZ, solid red line. Numbers in (**C**) indicate error probabilities (*p* values) as calculated with one-way ANOVA and Tukey multiple comparison test.

**Figure 9 cancers-14-06199-f009:**
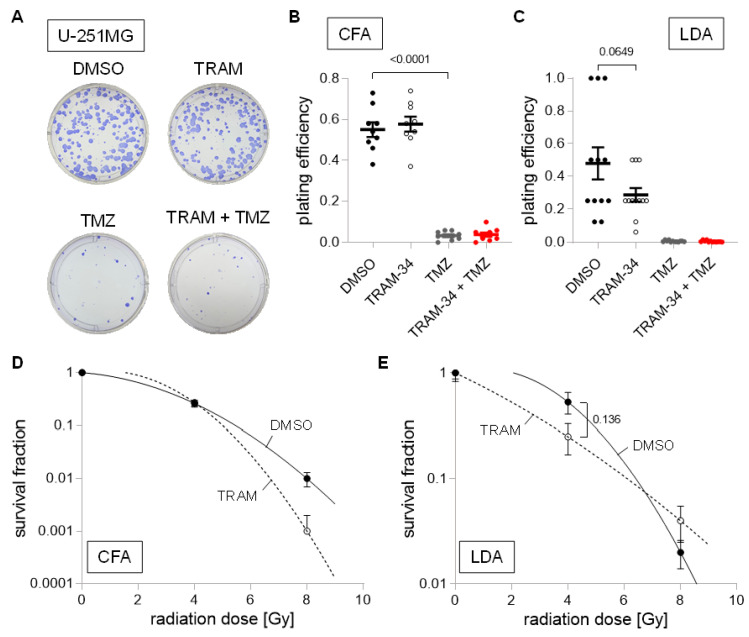
Cell culture condition-dependent effects of K_Ca_3.1 blockade on sensitivity to irradiation and temozolomide in U-251MG cells. (**A**–**C**). Representative images of colony formation (**A**) and plating efficiencies (**B**) of DMEM-grown or (**C**) NSC-grown cells after irradiation with 0 Gy and co-treatment with vehicle (DMSO, closed black circles), TRAM-34 (5 µM, open circles), temozolomide (TMZ, 30 µM, closed grey circles), or TRAM-34 (5 µM) together with temozolomide (TMZ, 30 µM, closed red circles). (**D**,**E**). Survival fractions of irradiated (0–8 Gy) cells co-treated with vehicle (DMSO, closed black circles) or TRAM-34 (5 µM, open circles) as determined by colony formation assay for DMEM-grown (**D**) and limited dilution assay for NSC-grown U-251MG cells (**E**). Data are individual values in (**B**,**C**) and mean ± standard error in (**B**–**E**) of three experimental units with three to four observational units each. Survival fraction curves were fitted according to the linear quadratic model and given as follows: DMSO, solid black line; TRAM-34, dashed line. Numbers in (**B**,**C**,**E**) indicate error probabilities (*p* values) as calculated with one-way ANOVA and Tukey multiple comparison test.

## Data Availability

The datasets used or analyzed during the current study are available from the corresponding author on reasonable request.

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
