# Peer review of "Tumoricidal, Temozolomide- and Radiation-Sensitizing Effects of KCa3.1 K+ Channel Targeting In Vitro Are Dependent on Glioma Cell Line and Stem Cell Fraction"

_cancers, 2022, doi:10.3390/cancers14246199_

Round 1

Reviewer 1 Report

In this study, the authors show that the blocking channel KCa3.1 led to inconsistent results regarding its temozolomide- and radiation-sensitizing effects, dependenting on glioma cell line and stem cell fraction. This is a potentially interesting study. However, there are several major points needed to be addressed: 

1.     Why did the authors test their hypothesis in mouse glioma cell lines but not in human cell lines? Human GBM cells and mouse GBM cells are not the same, or the authors should verify these results on human GBM cells.

2.     The authors should do western-blot to verify the qPCR results, sometimes they are not consistent.

3.     The authors should use flow to measure the expression of KCa3.1, which is a membrane protein.

4.     How long does it take to induce stem cells in NSG? And why cells are grown in 10 % CO2 atmosphere?

Reviewer 2 Report

The manuscript of Stransky et al. studies the effects of TRAM-34 (a KCa3.1 blocking agent) in different glioma cell lines (SMA-560, SMA-540, and GL-261) in temozolomide- and radiation sensitivity. The differences in cell lines and culture conditions result in heterogeneous results. So, the main aim of the manuscript is to caution about this aspect for future studies and to offer a possible explanation of the heterogeneity in patients with glioblastoma.

The article provides relevant information but there are some issues that should be addressed.

- Authors measure four stem cell markers (ALDH1A3, CXCR4, Nestin, and SOX2) and then, consider that all four have the same relevance to calculate a “stem cell score”. The relative relevance of each marker is unknown, therefore, this content (fig 1E-F) and their related conclusions should be removed.   

- It is difficult to see the effects of TRAM-34 in the GL-261 cell line because of the drastic effects that temozolomide (TMZ) has on this cell line. GL-261 cells die significantly in the presence of temozolomide, even without radiation (Figure 4SC and Fig 6B), then, the conclusions of the effect of TRAM-34 are not clear. Authors should indicate clearly that the effect of TRAM-34 in the GL-261 cell line, in the radiosensitivity with TMZ, couldn’t be addressed and eliminate their related conclusions.

Small changes:

Page 14, end of the second paragraph: *It may be correct to indicate Fig 6 instead:  “Furthermore, no effect of TRAM-34 was seen in GL-261 cells, arguably due to its low KCa3.1 expression compared to SMA-560 or SMA-540 cells (see Fig. 4). ”

Author Response

Point-to-point reply

Reviewer #2

- Authors measure four stem cell markers (ALDH1A3, CXCR4, Nestin, and SOX2) and then, consider that all four have the same relevance to calculate a “stem cell score”. The relative relevance of each marker is unknown, therefore, this content (fig 1E-F) and their related conclusions should be removed.   

            Author reply: We agree with the author and eliminated the stem cell score and changed the manuscript accordingly.

- It is difficult to see the effects of TRAM-34 in the GL-261 cell line because of the drastic effects that temozolomide (TMZ) has on this cell line. GL-261 cells die significantly in the presence of temozolomide, even without radiation (Figure 4SC and Fig 6B), then, the conclusions of the effect of TRAM-34 are not clear. Authors should indicate clearly that the effect of TRAM-34 in the GL-261 cell line, in the radiosensitivity with TMZ, couldn’t be addressed and eliminate their related conclusions.

Author reply: We agree that effects on radiosensitization are hard to analyze when the effects of TMZ, TRAM-34 (as is the case for U-87MG cells cultivated in NSC medium) or their combination are large when applied on their own. Hence, we added a cautioning paragraph and removed the analysis of radiosensitization from all graphs for all conditions showing large effects without prior radiation (which was also the case for the newly added U-87MG and U-251MG cell lines) to avoid confusions or wrong conclusions.

Small changes:

Page 14, end of the second paragraph: *It may be correct to indicate Fig 6 instead:  “Furthermore, no effect of TRAM-34 was seen in GL-261 cells, arguably due to its low KCa3.1 expression compared to SMA-560 or SMA-540 cells (see Fig. 4). ”

Author reply: We changed the paragraph and cite both Figures 3 and Fig. 7 (numeration changed due to additional patch clamp experiments).

Round 2

Reviewer 1 Report

Accept in present form

Reviewer 2 Report

The manuscript has improved substantially and the authors have answered all my previous questions. I recommend its publication.